# Prejudice Formation in Childhood: How Parental Bonding Can Affect Social Dominance Orientation

**DOI:** 10.3390/brainsci15111147

**Published:** 2025-10-25

**Authors:** Serenella Tolomeo, Shannen Koh, Gianluca Esposito

**Affiliations:** 1 Social and Cognitive Computing Department, Institute of High Performance Computing, Agency for Science, Technology and Research, Singapore 138632, Singapore; serenella_tolomeo@a-star.edu.sg; 2Psychology Program, School of Social Sciences, Nanyang Technological University, Singapore 639818, Singapore; 3Department of Psychology and Cognitive Science, University of Trento, 38068 Rovereto, Italy

**Keywords:** amygdala, Social Dominance Orientation, Parental Bonding Instrument, parental bonding styles, grey matter volume

## Abstract

**Background:** How individuals develop and form perspectives of those around them differs from person to person. Factors such as childhood parental bonding styles can affect how prejudice forms. Social dominance in adulthood may also be affected by childhood experiences through the bonding received. Not many studies examine how an individual’s Social Dominance Orientationcan be influenced by parental bonding styles in childhood. Furthermore, few studies that investigated neural correlates are associated with these two variables. As such, this study aims to establish how parental bonding in childhood affects brain regions that are also implicated in adult SDO. **Methods:** Ninety-one participants were recruited and underwent magnetic resonance imaging (MRI). Social Dominance Orientation (SDO) and Parental Bonding Index (PBI) were collected. We used DARTEL package in SPM12 to conduct a whole-brain analysis. The ROI analyses were focused on amygdala grey matter volume (GMV). **Results:** This study identified a strong correlation between PBI and SDO. Interestingly, PBI_Care_ and PBI_Protection_ scores significantly predicted SDO scores. SDO was positively associated with amygdala GMV, PBI_Care_ was negatively associated with amygdala GMV, and PBI_Protection_ was positively associated with amygdala GMV. **Conclusions:** Our results show that PBI and SDO are highly correlated as well as their association with the amygdala and other key regions of the brain.

## 1. Introduction

Humans are a social species that flourish in collaboration with others [1]. Forming groups and hierarchies are part of our nature and dominance hierarchies within these systems translate to an unequal allocation of resources to members of differing ranks [2]. Competition is henceforth inevitable and perceived as a common aspect of social systems [3]. In particular, adolescence is a critical developmental period where there is increased neural plasticity toward external experiences, and where the formation and development of belief systems and individual identity occur [4,5]. As such, as these individuals mature, their developing cognitive and social skills provide cues on their role within social groups [4]. Research on development has found that family processes have a notable contribution to the formation of social attitudes and subsequently, prejudice towards others [6,7].

Social Dominance Orientation (SDO), a system theorized to develop during childhood, addresses how individuals may display prejudice toward the outgroup and endorse the maintenance of intergroup inequality [7,8,9]. Such social goals can be explained by the transference of parental social attitudes to their offspring during sensitive periods in development [10,11]. Parental influences during these periods of increased sensitivity can thus cultivate the resistance or acceptance of group identity. As SDO encourages group-based dominance, lower-ranking members begin doubting their identity and provide justifications for their inequalities, which ultimately leads to no changes in their circumstances [7]. The individual level of SDO is thus an essential component in the formation of social identity and intergroup attitudes [7]. Despite studies being conducted on SDO, there has been no clear consensus on its developmental process. Nonetheless, studies have suggested that SDO development is guided by both parental influences and the child’s personality [12].

Parental bonding styles contribute greatly to brain development and the maturation of neurological systems that lead to normative emotional and behavioural regulation [5,13,14]. Parental bonding styles can be measured using the Parental Bonding Instrument (PBI). It is a self-report measure, based on a 25-item questionnaire that determines parental styles as perceived by the individual during their first 16 years [15]. The scale comprises 2 subscales; Parental Care (PBI_Care_), and Parental Overprotection (PBI_Protection_). Increased brain plasticity during adolescence means that individuals have a greater susceptibility to environmental and external influence [16,17]. A study by Hawley [18] posits that socializing experiences in early life essentially sets the path the child would follow. For which parents are usually key figures that shape the child’s experiences [5,19]. In the same vein, early experiences of insensitive caregiving styles were found to have an acute impact on social functioning, including how children manage social interactions that are related to dominance [3].

Social dominance in individuals can be measured through the Social Dominance Orientation (SDO), which is a measure that evaluates the extent to which one seeks intergroup hierarchy [20]. SDO is related to the desire for intergroup inequality and hierarchical maintenance, and can reveal one’s intrinsic beliefs about social hierarchy [4]. SDO is essentially a construct disclosing intrapersonal attitudes towards intergroup relations, where people scoring high in SDO exhibit preference for their social group and demonstrate prejudice towards outgroups [20]. Moreover, individuals high in SDO perceive the world through a competitive lens where social groups fight for scarce resources [10]. In adolescents who are more socially dominant, the concept of segregation based on dominance is acceptable and justified [12]. Adult SDO scores were also found to positively correlate with prejudicial pro-ingroup and anti-outgroup attitudes [4,9,21].

The observation that all existing human communities consist of group-based hierarchies can be explained by the social dominance theory (SDT) [22] which postulates that group dominance is determined by the group that possesses high-value resources. Their position is therefore maintained through complex interactions between both parties who share an agreement about the control of these resources [4].

For example, bullying is a common form of dominant expression. Repeated forms of aggression (relational or physical) towards another would clearly illustrate a power differential among peers in terms of social status, which could promote the development of SDO [18]. Bullying others indicate the use of coercive resource control strategies, where those who are more social-dominance oriented benefit through social rewards such as popularity or more friends [18]. SDO has been reliably proven to be consistent with stable negative attitudes towards outgroups [10].

It has been established that SDO and its development is largely an internal process that is influenced by experiences during childhood [4]. One’s family is considered an important factor in the development of prejudice [10]. Specifically, parents and close relatives can inculcate SDO in children by teaching them that gaining dominance at the cost of other groups is unavoidable [8]. Parental bonding styles classified as “unaffectionate socialization”, wherein parents provide inconsistent or little warmth and responsiveness was seen to be correlated with SDO scores [9,23]. Research suggests that strict parenting styles are directly related to the child’s endorsement of dominance goals in reactance to ambiguous stimuli [24]. This occurs especially if parents emphasize the achievement of extrinsic goals (i.e., status and power) more than intrinsic goals during childhood [25]. Inclination towards such goals increases aggressive attitude and encourages prejudice formation towards the outgroup, increasing SDO [11]. Furthermore, a study by Duriez et al. [26] found significant parent–child SDO correlations. Parents who were perceived to engage in positive parenting behaviours and were more responsive had children with lower SDO scores, with responsiveness composing 4% of the variance in SDO scores [4,11]. The effects of SDO transference between parent to child are further exacerbated when familial attitudes are congruent, which supports the view that parents are significant influencers on SDO development in children, persisting till adulthood [10].

In spite of neurological studies on social interactions being conducted, not much is known about the neural basis that makes up social dominance and relationships related to it. One region of the brain that overlaps in relation to parental styles and SDO is the amygdala. A voxel-based morphometry (VBM) study by Kumaran et al. (Figure 1a) [27] found that individual differences in amygdala grey matter volume (GMV) are related to learning of social hierarchies, with a positive correlation between SDO and amygdala GMV [28]. To date, there has been no defining study that has found brain regions accounting for social dominance independently [28]. Rather, it is a network of brain structures that determine SDO in an individual [28].

With regard to GMV in this aspect, early deprivation of nurture and lack of parental sensitivity led to reductions in grey and white matter volume in adults. Contrastingly, adults who reported nurturing parental care and parental sensitivity in childhood exhibited larger GMV and total brain volume compared to controls [29]. In summary, a variety of studies has revealed brain structures associated with social dominance and parental bonding styles. Namely, the prefrontal cortex, anterior cingulate cortex (ACC), amygdala, and GMV in the amygdala and dorsomedial prefrontal cortex (dmPFC). However, this study only focused on how differing bonding styles affect amygdala GMV.

Multiple studies have found that subjects who were maltreated during childhood displayed hyperresponsiveness of the amygdala [30,31,32,33]. Neuroimaging research conducted on youths supported this claim with results showing that high levels of negative maternal behaviours were associated with increased amygdala activation (Figure 1b) [9,15]. However, there were no references as to how activity in the amygdala may affect its relative size. In a longitudinal study by Kok et al. [29], maternal responsiveness predicted reduction in amygdala growth and greater thinning of the orbitofrontal cortex over a period of 4 years. That is, when an individual receives better parental care, there is a decrease in amygdala growth [29]. As such, this study aims to determine if the PBI can affect amygdala GMV.

Larger amygdala corresponds to better performance in learning about social hierarchies, when the development of the amygdala is hindered, this prevents the individual from learning how to evaluate social hierarchies, which may lead to diminished SDO [34]. In line with dominance affecting structural changes in the brain, studies that examined the effect of psychopathy on amygdala volume found a negative association between total psychopathy scores and left amygdala volume [35]. Hence, we hypothesize that SDO is positively associated with amygdala GMV.

Dominance is generally classified with intrinsic factors such as genetics or personality. However, how dominance is conveyed depends more on behavioural models such as peers, parents, and close relatives [3]. Across different cultures and nations, studies suggest that traditional families, in which the family structure is patriarchal, produced more children with higher SDO, whereas children from more egalitarian families had lower SDO scores in comparison. In a setting such as Singapore, many families follow the ideal family structure in which the patriarch has unquestioned authority [36]. There also remains a paucity of research examining SDO in Singapore. SDT is a concept that demonstrates the good potential to produce novel revelations in Singapore with the inclusion of SES and gender-egalitarianism in family structures and how they can affect SDO in adulthood. However, the complex interactions between parental bonding styles and SDO have not yet been thoroughly investigated in the context of an Asian population such as Singapore.

Even though particular brain regions have been found to be associated with social dominance, not much research has been conducted on the neural links between parental bonding styles and SDO. Hence, this paper aims to establish how parental bonding styles in childhood affects regions of the brain that are also implicated in SDO. Measures that determine an individual’s SDO and perceived parental bonding were collected through self-report scales. This examination of the neural link between these two concepts may shed new knowledge on the influence of parental bonding styles on the development of attitudes and prejudice towards others on the neurological level, identifying any existing neural correlations between parental bonding styles and SDO. Results may also indicate how parental bonding behaviour in early childhood may significantly influence regions of the brain that are associated with social dominance, affecting SDO in adulthood. We tested the following five hypotheses: (1) PBI and SDO demonstrate a significant correlation; (2) PBI_Care_ and PBI_Protection_ scores serve as significant predictors of SDO scores; (3) SDO shows a positive association with amygdala GMV; (4) PBI_Care_ demonstrates a negative association with amygdala GMV; and (5) PBI_Protection_ demonstrates a positive association with amygdala GMV. To test these hypotheses, we conducted Pearson’s correlation analysis and multiple linear regression analysis.

## 2. Methods

### Participants

Ninety-one (46 females, age range = 21–42) participants were recruited for this study in 2 phases. 56 (27 females) participants (M = 23.05, SD = 1.30) recruited for the first phase were Singaporean Chinese students from Nanyang Technological University (NTU). In the second recruitment phase, identical inclusion and exclusion criteria were used. The only difference for this phase was the target sample, which was middle-aged adults from the community. There were 35 (19 females) participants (M = 31.59, SD = 6.48) in the second phase. Participants aged 21 and above were selected to ensure stable retrospective reporting of childhood parental bonding experiences (as measured by the PBI for the first 16 years of life) and to avoid potential confounds from ongoing adolescent neurodevelopmental processes that could affect both self-report accuracy and brain structure measurements.

Participants were included based on the following criteria: (a) English speaking, (b) Chinese ethnicity, (c) no diagnosis of intellectual disabilities, (d) right-handed, (e) normal or corrected-to-normal vision and hearing, (f) no psychiatric/neurological illness, (g) no history of illicit drug use, and (h) had not travelled overseas for more than two months over the six months from the time of the scan. Exclusion criteria included: (a) history of psychological or neurological disorders, (b) pregnancy, (c) left-handedness or ambidexterity, (d) uncorrected vision or hearing impairments, (e) current or past substance use (alcohol, nicotine, caffeine within 24 h prior to session; illicit drugs at any time), and (f) extended overseas travel (>2 months in the 6 months prior to scanning). Each participant was required to provide written informed consent before participating in the study. This study was conducted according to the guidelines of the Declaration of Helsinki, and it was approved by the Institute Review Board of the Nanyang Technological University.

## 3. Measures

### 3.1. PBI Scale

The Parental Bonding Instrument (PBI) [15] is a scale that measures parental bonding styles during the first 16 years of life (see Appendix A). It is a 25-item retroactive self-report in which 2 domains of parental bonding styles, parental care and parental overprotection were evaluated. Of the 25 items in the questionnaire, 12 items measure ‘care’ and the remaining 13 items measure ‘overprotection’. Items are scored on a 4-point Likert scale (0 = very unlike, 3 = very like) with some items reverse-coded. The scale used in this study included statements like, “Appeared to understand my problems and worries”, “Enjoyed talking things over with me” which fell under the Care Subscale, and “Tried to control everything I did”, “Invaded my privacy” which fell under the Protection Subscale. Scores derived from this scale would classify parents into one of the following four parental bonding quadrants: “affectionate constraint” = high care and high protection, “affectionless control” = high protection and low care, “optimal parenting” = high care and low protection, and “neglectful parenting” = low care and low protection (see Table 1). The cut-off scores used to distinguish “high” and “low” categories were derived from the original PBI validation study [15] and replicated in subsequent validation work [37], based on normative population data. The PBI has been shown to be a psychometrically sound instrument, with high test–retest reliability for the care scale (*r =* 0.76) and the protection scale (*r =* 0.63) over a three-week interval [37]. In this study, PBI scores were acquired from the sum of each subscale, taking into account the reverse-coding of some items and analyzed as a continuous variable.

### 3.2. SDO Scale

Social Dominance Orientation (SDO) [20] is a self-report assessment used to measure social dominance in individuals (see Appendix B). The SDO scale has been shown to have good internal reliability and validity [20]. The measure consists of 16 items, rated on a 7-point Likert scale (1 = strongly disagree, 7 = strongly agree). Some statements in the SDO include, “It’s probably a good thing that certain groups are at the top and other groups are at the bottom”, “Groups at the bottom are just as deserving as groups at the top”, and “We should not push for group equality”. SDO scores were derived from the sum of participants’ responses to each item, after reverse-coding and analyzed as a continuous variable.

### 3.3. Self-Report Data Analysis

Cronbach’s alpha was calculated for both the PBI and SDO to determine inter-item reliability. Any existing correlation between PBI and SDO was evaluated using Pearson’s correlation. Additionally, a multiple linear regression was carried out to determine if PBI could effectively predict SDO. The stated statistical tests were performed with SPSS 23.

### 3.4. MRI Acquisition

High-resolution T1-weighted MPRAGE sequences (192 slices; TR 2300 ms; TI 900 ms; flip angle 8 degrees; voxel size 1 mm) were collected from a Siemens Magnetom Prisma 3-Tesla MRI Scanner with 64-channel head coil. All participants were fitted with a head restraint to prevent excessive head movement during the scan. No participant was found to have abnormalities in their brain structures.

### 3.5. VBM Pre-Processing

Structural data were processed using Statistical Parametric Mapping (SPM12; Wellcome Department of Imaging Neuroscience, http://www.fil.ion.ucl.ac.uk/spm/software/spm12 (accessed on 1 June 2021)) on Matlab 2021a platform. First, T1-weighted images were segmented using the diffeomorphic anatomical registration through exponentiated lie algebra for the segmentation (DARTEL) and normalization with modulation to preserve the total amount of grey matter, then transformed into the Montreal Neurological Institute (MNI) stereotactic space to produce 1 × 1 × 1 mm^3^ voxels. Finally, they were smoothed by convolving the images with an isotropic Gaussian kernel of 8 mm full width at half maximum (FWHM) [38].

### 3.6. Whole-Brain Analysis

We used the DARTEL package in SPM12 to conduct an exploratory whole-brain analysis. PBI or SDO scores were used as contrasts to test significance of regression coefficients from the zero value. Significance thresholds were set at an uncorrected voxel-wise level of *p* < 0.01.

In these analyses, we aim to control for total intracranial volume (TIV), age, and gender by including them in the regression model as independent ‘nuisance’ variables. The inclusion of TIV as a variable to account for is particularly important in ROI-based volumetric neuroimaging measures because slight differences in a specific region of the brain may become confounded with individualistic total brain size [39].

We are controlling for age not just because TIV changes with age [40], but also because perceptions of parental behaviour change as the individual grows older, with parents perceived as less caring by older children [41]. Parental overprotectiveness in particular, was perceived more negatively by adults than children [42]. Accounting for age is also critical in this study as the sample includes participants from two different age groups, a middle-aged sample and a young-adult sample. Differences in the self-report measures and in their total brain size is expected and by including age into the regression model, it reduces confounds from existing age differences.

Lastly, past studies have also suggested a gender difference in self-reports of PBI and SDO. Specifically, men tend to have higher SDO scores compared to women, even across ethnic groups [20,43]. Gender differences in SDO were also found to be largely invariant across cultural, situational, and contextual thresholds [43]. In contrast, women tend to score higher on PBI_Protection_ and lower on PBI_Care_ compared to men [44]. Along with an overall gender difference in total brain size between men and women [45,46], controlling for gender would aid in identifying neuroanatomical correlates, similar to the age variable. However, although PBI, SDO, and total (and regional) brain size are likely to vary with respect to age and gender, these two variables were not included in the main analysis and were treated as nuisance variables.

### 3.7. ROI Analysis

The ROI analyses were focused on: (1) amygdala GMV was positively associated with both SDO and PBI_Protection_ scores, and (2) amygdala GMV was negatively associated with PBI_Care_ score. The key independent variables are the two possible determinants of prejudice formation in childhood (i.e., PBI and SDO scores). For each participant, an average voxel-wise GMV value was determined for each ROI individually, which then serves as the dependent variable for our main analyses.

We plan to investigate the association between GMV and scores in the PBI and SDO scales using an ROI multiple regression analysis. Predictors were included SDO, PBI_Care_ and PBI_Protection_, with TIV, age, and gender as covariates. As the brain areas in our hypotheses are identified with a strong a priori prediction, the threshold of significance was set at uncorrected voxel-wise level of *p* < 0.01. To analyze these regression models, we were using the MarsBaR toolbox 0.44 (http://marsbar.sourceforge.net/ (accessed on 7 September 2025)) to anatomically define ROIs for the amygdala, as per our main hypotheses. The GMVs were extracted from their ROIs using anatomically defined spheres with a radius of 20 mm centred at (MNI: x = −28.5, y = −6, z = −5) for the left amygdala and at (MNI: x = 21, y = 4.5, z = −18) for the right amygdala [47].

## 4. Results

### 4.1. Self-Report Data

A preliminary analysis was conducted for both the PBI and SDO scales to ensure that both measures satisfied the assumptions required for parametric testing. All items in both the PBI and SDO did not exceed the recommended cut-off values of an absolute 3 (±38). Hence, all items were deemed normally distributed and were included in the following analyses. Additionally, both the PBI and SDO exhibited acceptable and excellent internal reliability at α = 0.48 and α = 0.91, respectively. A correlation analyses was then conducted to determine the extent of correlation between the PBI and SDO. Table 2 illustrates the correlations between the self-report variables used in this study. It can be noted that there is a weak correlation between the SDO and the PBI (*r* = −0.19, *p* < 0.001). Furthermore, correlation analyses between the PBI_Care_ and PBI_Protection_ subscales to the SDO also revealed a poor Pearson’s Correlation (*r* = 0.15, *p* < 0.001; *r* = −0.11, *p* < 0.001), respectively.

Next, a multiple linear regression was conducted to test if PBI_Care_ and PBI_Protection_ scores significantly predicted SDO scores. The fitted regression model was: SDO = 88.42 + 0.182B_1_ − 0.11B_2_ (where B_1_ = PBI_Care_ B_2_ = PBI_Protection_). The overall regression was not statistically significant (R^2^ = 0.03, F(2,78) = 1.08, *p* = 0.35). It was found that PBI_Care_ could not significantly predict SDO (*β* = 0.13, *p* = 0.28), and PBI_Protection_ did not significantly predict SDO as well (*β* = −0.06, *p* = 0.62).

In order to further examine the correlation between the PBI and SDO, we were continuing with the whole-brain analysis and ROI analysis as described earlier. This was conducted to determine the presence or absence of any neural correlates between parental bonding and social dominance. Prior to that, an independent-samples t-test was run to determine if there were any differences in the self-report scores between males and females. There were no gender differences between scores for the SDO (*t* = −1.19, *p* = 0.24) and PBI_Protection_ (*t* = −0.25, *p* = 0.80); however, a significant gender difference was found for PBI_Care_ scores (*t* = 2.33, *p* = 0.02). As such, gender was included in the VBM analyses as a covariate as mentioned in the methodology.

### 4.2. Whole-Brain Analysis

For the main analysis, we ran a whole-brain analysis with PBI and SDO as variables of interest. TIV, age, and gender are also included into the analysis model as covariates to regress out any potential effects. As this analysis was exploratory, there were no a priori hypotheses to test. A significance level of *p* < 0.01 and an extent threshold of 20 contiguous voxels was used for this study which indicates a sphere radius of 20 mm. This threshold was selected given the exploratory nature of the analyses, to reduce the likelihood of Type II errors while still requiring a cluster-extent criterion. We note that this choice increases the risk of Type I errors (false positives), and therefore results should be interpreted with caution. All *p*-values reported in this study are uncorrected *p*-values.

To begin, we looked at which regions of the brain were associated with SDO, followed by regions associated with PBI. Our analyses revealed that SDO was positively associated with multiple areas in the brain such as the left frontal motor cortex (*P_uncorr_* = 0.004, MNI: x = −22, y = 3, z = 63), and right frontal motor cortex (*P_uncorr_* = 0.003, MNI: x = 16, y = −3, z = 62). Additionally, significant clusters were also found in the right fusiform gyrus (*P_uncorr_* = 0.002, MNI: x = 56, y = −46, z = −16) and the right amygdala (*P_uncorr_* = 0.007, MNI: x = 44, y = −42, z = 2).

Thereafter, we directed our analyses to regions of the brain associated with PBI. At a significance level of *p* < 0.01, the PBI was significantly associated with multiple parts of the brain as well. One area in particular was the significant cluster in the parietal lobe (*P_uncorr_* = 0.004, MNI: x = −3, y = −48, z = 63) that was associated with both subscales at similar MNI coordinates. Our analyses also found that the PBI_Care_ has significant clusters in the left anterior prefrontal cortex (*P_uncorr_* = 0.004, MNI: x = −14, y = 46, z = −4), the left dorsal posterior cingulate cortex (*P_uncorr_* = 0.001, MNI: x = −16, y = −48, z = 34), and in the left amygdala (*P_uncorr_* = 0.001, MNI: x = −8, y = 38, z = −20). Unexpectedly, the whole-brain analyses also suggested that there are brain regions that were associated with both SDO and PBI. There were significant clusters found in brain areas associated with PBI and SDO that overlap. First, an association of the PBI_Care_ to the SDO in the motor cortex (*P_uncorr_* = 0.001) and an overlap between SDO and PBI_Protection_ with a significant cluster in the fusiform gyrus (*P_uncorr_* = 0.001).

### 4.3. ROI Analysis

In addition to the whole-brain analysis, an a priori ROI analysis was also conducted with the PBI and SDO, with the same covariates included to regress out any potential effects. As per the whole-brain analyses, a threshold of 20 voxels was set as the criteria to determine any significant clusters. The amygdala’s MNI coordinates (left amygdala: −28.5, −6, −15; right amygdala: 21, 4.5, −18) were retrieved from a similar-design study by Suffren et al. [47] and used in the analysis with a spherical radius of 20 mm to determine ROI margins. SDO was positively associated with GMV in a cluster within 20 mm radius of the left amygdala (*P_uncorr_* = 0.09, MNI: x = −46, y = −2, z = −20), and in the right amygdala (*P_uncorr_* = 0.16, MNI: x = 38, y = 14, z = −14). Subsequently, we focused on the association of the amygdala with PBI_Care_ and PBI_Protection_ separately. Results found that both PBI_Care_ and PBI_Protection_ were associated with the right amygdala. However, the PBI_Care_ was identified to be associated with the left amygdala as well. Specifically, the PBI_Care_ was found to have a negative association with both the left amygdala (*P_uncorr_* = 0.006, MNI: x = −45, y = −15, z = −10), and the right amygdala (*P_uncorr_* = 0.001, MNI: x = 34, y = 18, z = −20). In contrast, only one positively associated significant cluster was found for the PBI_Protection_ in the right amygdala (*P_uncorr_* = 0.004, MNI: x = 30, y = 20, z = −27). Table 3 summarizes all clusters in the amygdala that were associated with PBI and SDO.

## 5. Discussion

This study sought to identify the presence of correlations between the PBI and SDO, and to determine the direction of associations between these two constructs with the amygdala. To begin, we conducted correlation analyses on the PBI and SDO constructs to determine if there were underlying correlations between them. However, we found a weak negative correlation between the SDO and PBI. The results of another correlation analysis found weak correlational relationships between the subscales and SDO. A multiple linear regression also found that PBI_Care_ and PBI_Protection_ scores could not significantly predict SDO scores. However, these results do not necessarily mean a non-significant relationship on the neurological level. As such, an exploratory whole-brain analysis and ROI analyses were conducted to further establish the presence of any neurological associations between PBI and SDO. We reported substantial overlap in brain activation patterns in regions involved in emotion, social cognition and bonding. Findings demonstrated a positive association between the SDO with the bilateral frontal motor cortex, and fusiform gyrus, and a negative association with the right amygdala. Table A1 in Appendix C provides an overview of the study’s hypotheses, analyses, and the degree to which findings supported them.

With respect to the PBI, both subscales were found to be associated with the parietal lobe. Additionally, PBI_Care_ was negatively associated with the left anterior prefrontal cortex, left dorsal posterior cingulate cortex, and the left amygdala. Additionally, results from the whole-brain analysis revealed that there were regions of the brain that were linked to both PBI and SDO. Significant clusters were found in the motor cortex, and the fusiform gyrus. Next, an ROI analysis was conducted to verify any neurological associations present. The results indicate that there is a significant neurological association between the SDO and the amygdala. SDO was found to be positively associated with amygdala GMV bilaterally. Next, our analyses focused on the associations between the PBI_Care_ and PBI_Protection_ subscales to the amygdala. The data suggest that there is a negative relationship between PBI_Care_ and amygdala, with significant clusters found bilaterally on the amygdala. Lastly, PBI_Protection_ only exhibited a significant cluster on the right amygdala and not the left.

It is important to note that while neuroimaging analyses revealed significant associations between PBI, SDO, and amygdala GMV (supporting Hypotheses 3–5), the self-report correlational analyses did not support Hypotheses 1 and 2. Specifically, we found weak correlations between PBI and SDO at the behavioural level, and PBI subscales did not significantly predict SDO scores in the regression analysis. This discrepancy between behavioural and neural findings warrants careful consideration. Several explanations may account for this divergence. First, self-report measures may be limited by recall biases, social desirability effects, or restricted variance in this relatively homogeneous sample [48,49]. Second, neural correlates may reflect underlying neurobiological processes that precede or exist independently of conscious attitudes captured by questionnaires. Third, the relationship between parental bonding and social dominance may be mediated by neural mechanisms that are not directly detectable through behavioural correlation alone. This pattern suggests that a multi-level analytical approach (i.e., combining self-report and neuroimaging data) is essential for understanding the complex relationships between early parenting experiences and adult social attitudes. While our neural findings suggest meaningful associations at the brain level, the lack of strong behavioural correlations requires that we interpret our conclusions with appropriate caution and acknowledge that the relationship between PBI and SDO may be more nuanced than initially hypothesized.

Our current findings suggest that the amygdala is one region associated with social dominance. This part of the brain, located bilaterally in the medial temporal lobe, is an essential brain structure used to assess and evaluate hierarchical social systems [50]. When parental bonding styles are not exemplary throughout development, this modifies normal cognitive development and causes its acceleration. The stress acceleration hypothesis (SAH) [51] propounds that negative experiences lead to increased brain development that is adaptive in the short term but maladaptive in the long run. This escalation affects emotional and behavioural regulation in adulthood by reducing neuroplasticity [51]. Studies have suggested that amygdala damage affects social functioning in humans, and also our primate relatives. A study on rhesus macaques that were subjected to extensive damage to their amygdala plunged to the bottom of a dominance hierarchy compared to their conspecifics [52]. Similarly, amygdala-lesioned monkeys seemed to exhibit more fear-response behaviours, and demonstrated less species-typical aggression toward their non-lesioned counterparts [53]. In humans, a larger bilateral amygdala GMV was found to be related to higher transitivity performance in learning about a traditional hierarchical system [50].

Activity in the amygdala and striatum, a substructure in the limbic system, significantly decreased when exposed to social defeats. Our findings of a positive association between the SDO and amygdala support past functional neuroimaging studies suggesting that the amygdala is essential for navigating social dominance. Multiple brain areas were revealed to be associated with the PBI from our analyses. Notably, the parietal lobe exhibited significant clusters when whole-brain analyses were conducted. Both PBI_Care_ and PBI_Protection_ were linked with activity in the parietal lobe, and can be explained by the neurological basis of human attachment. When mothers observed their own child as compared to an unknown child, their inferior parietal cortices were activated [54]. This area of the brain is typically classified with perception-action theory, including visual-association areas and mirror neuron system areas, possibly activated by the mothers’ observing their child [54]. The same study also suggested that the bilateral amygdala and left cingulate cortex are brain structures related to maternal bonding. In addition, lower brain activity in the left amygdala and posterior cingulate cortex (PCC) was observed when attachment level is high [55]. The simple act of bringing to mind mothers as compared to close friends accounted for activity in the prefrontal cortex (PFC) and anterior cingulate cortex as well [55,56]. Moreover, the orbitofrontal cortex (OFC), part of the PFC, has been posited to be a significant region of the ‘social brain’ used for processing social cues and may be involved in human parenting behaviours [57].

In the classic study by Eisenberger et al. [58], perceived social exclusion from an online game led to rapid changes in the participant’s cingulate cortex. Leading researchers to believe that the cingulate cortex is a mediator for social loss and attachment, which are key aspects of parenting and the development of a child. Although these studies were mainly neuroimaging on mothers’ brains, these results may mirror that of people who are engaged in retrospective recall of parental care and concern during childhood. As such, the brain regions mentioned may demonstrate a change in activity levels when tasks similar to maternal bonding are performed [54]. These findings advocate for why the PBI may activate these specific brain regions in our participants.

Despite the lack of studies that investigate the neurological correlates between parenting bonding and dominance, our study suggests that there may be potential correlations between these two constructs. Specifically, we found significant overlaps between PBI and SDO in two parts of the brain—the fusiform gyrus and the motor cortex. Mothers who exhibit high maternal care for their children during childhood showed larger GMV in their fusiform gyrus, and in other parts of their brain. Similarly, when attachment levels are high, reduced brain activity was observed in the putative occipital face area (OFA), which is part of the fusiform gyrus [55]. This brain structure is also activated during memory tasks, with imaging data illustrating signal increases in the left fusiform gyrus when participants are engaged in recall [59]. Concerning the motor cortex, an extended search online revealed no significant studies on the correlation between PBI and SDO. Perhaps future research can investigate if the motor cortex is associated with PBI and SDO on a neuropharmacological level instead.

Our ROI analyses focused on the amygdala and how PBI and SDO affect amygdala GMV. Our findings suggest that SDO and PBI_Protection_ were positively associated with amygdala GMV, whereas PBI_Care_ was negatively associated with amygdala GMV. There has been a long-standing view that the amygdala is a key structure catered for instinctual evaluation of incoming stimuli that may threaten one’s wellbeing [60,61]. As such, a perceived threat to one’s social dominance may activate the amygdala [61]. It was found that decreased activation of the amygdala with greater attachment may be linked to the amygdala processing incoming fear/safety signals [55,62]. These studies provide further reinforcements for the positive associations found between SDO and amygdala GMV in our ROI analysis. The amygdala has also been linked to social affiliation and maternal attachment [63,64]. In fact, children with depressed mothers show greater proclivities to deviant social behaviour, dysfunctional stress management, and importantly, exhibit larger amygdala volume [63]. This supports the notion that amygdala GMV has a negative relationship with the PBI_Care_ subscale, in line with our study’s ROI analysis. Perhaps a poor parental relationship during one’s childhood may translate into larger amygdala GMV after maturity. As seen in our results, a higher PBI_Care_ is associated with smaller amygdala GMV, in consonance with existing literature. A greater PBI_Protection_ score typically indicates a more ‘constraining’ or ‘controlling’ parental bonding style [15]. However, it is important to note that the overprotection subscale can also capture elements of anxious parental involvement, where heightened concern and monitoring may not always reflect purely negative control. This dual nature of the construct suggests that elevated PBI_Protection_ scores could represent both protective concern and restrictive behaviours. Such parental involvement may influence children’s emotional regulation and stress responsivity differently, which could help explain the observed positive association with amygdala GMV. In turn, these neural correlates may contribute to the formation of Social Dominance Orientations in nuanced ways, depending on whether the child internalizes parental overprotection as supportive vigilance or restrictive control.

The role of neurochemicals in parental bonding and social dominance may play a part in parenting behaviours or how we establish social hierarchies. Serotonin is a neurotransmitter that has been shown to play a major role in evaluating social rankings. For instance, enhancement of serotonin signalling in a group of velvet monkeys led to increased dominance, and vice versa. Another neurotransmitter, oxytocin, works similarly by facilitating maternal behaviours and promoting nurture and responsivity [65]. Subsequently, the role of such neurotransmitters should be thoroughly investigated in future studies to determine if their concentration in childhood plays a part in the formation of social dominance.

Limitations exist with our study as well. Firstly, the cross-sectional nature of the design does not allow us to determine whether the observed results relate to formation or maintenance of the neural links between PBI and SDO. Secondly, while we found significant neural associations between PBI, SDO, and amygdala GMV, the behavioural correlations between these constructs were weak and non-significant. This discrepancy suggests that self-report measures alone may not fully capture the complex neurobiological relationships underlying parental bonding and social dominance. Future research should employ multiple assessment methods and larger sample sizes to better understand this divergence between behavioural and neural levels of analysis. Thirdly, our sample population also included a rather homogenous group of participants—all Chinese participants. This homogeneity increased our power to detect any neurological correlates, but in return, we compromised on our ability to detect if these relationships may be a result of cultural effects. Parenting practices and their interpretation are embedded within cultural norms. For example, patriarchal family structures are particularly salient in Singaporean Chinese families [36]. As such, the meaning of parental bonding behaviours (e.g., protection vs. control) may differ across cultural contexts, and our findings may not be directly generalizable to Western or other populations. Future research should therefore aim for explicit cross-cultural replication, with more diverse samples, to clarify whether the observed associations hold across different cultural settings or are specific to collectivistic, patriarchal family systems.

Next, the PBI is a self-report scale that utilizes retrospective recall [15] and the participants’ rating on the PBI may be influenced by their current mental state or life experiences. However, studies have repeatedly established that the PBI has strong psychometric properties, and that current mood were not biassed results [15]. Although this may bolster support for the PBI, future researchers who plan to use the PBI would not be disadvantaged if they requested family or medical history from their participants beforehand. It is also important to acknowledge that the PBI Overprotection subscale may capture a dual construct, encompassing both anxious parental concern and restrictive control. This conflation of protective vigilance with constraining behaviours may limit interpretive clarity, and future studies could benefit from using more fine-grained measures that distinguish between supportive involvement and controlling parental styles. Finally, we note that the statistical threshold applied in our analyses (uncorrected voxel-wise *p* < 0.01 with a 20-voxel cluster extent) is lenient and increases the risk of false positives. While this choice was made to balance sensitivity and specificity in an exploratory context, future studies should employ more stringent correction methods (e.g., FWE or FDR correction) to validate these findings.

Future studies can explore the same hypotheses in notable Singaporean studies such as GUSTO. GUSTO is a longitudinal study that examines how conditions during pregnancy and early childhood influences the subsequent health and development of women and children. Using GUSTO as a starting point would be beneficial as its sample pool includes mothers from all ethnicities and backgrounds. Including participants with multi-cultural backgrounds may help negate any potential cultural effects that may confound the results. Additionally, a longitudinal design that includes the influence of genetics and other environmental factors may be beneficial in further confirming that PBI can affect SDO and its relative brain regions in the long-term. Future studies are encouraged to take reference from the design style of GUSTO to conduct longitudinal studies on the PBI and its impact on SDO in adults.

Next, research has found that nuclear family units with a patriarchal familial structure produced significantly more children who score higher on the SDO [36]. In Singapore, where most families have been observed to parallel the patriarchy, perhaps future studies may seek to identify if the type of familial system has any significant impact on SDO. Lastly, most of the studies found in the literature on parental bonding involved mothers’ and their children. There seem to be few studies that focus on how a father’s role in parenting may play a part in their child’s development and subsequent SDO formation. In fact, the lack of paternal care in animals was found to affect synaptic development in the anterior cingulate cortex [66]. The presence of a father increased chances of offspring survival and also aided in increasing maternal behaviour [67]. This has shown that the effect of paternal care and involvement on the developing mind has been duly underestimated. Hence, future studies can investigate the effects of paternal care on the mind of the developing child, and how early paternal behaviours may affect adulthood. Besides that, perhaps the reverse relationship could be explored in future studies. Such as how an individual’s rating on the SDO may affect their future parental bonding style.

Our limitations notwithstanding, our current findings further expand the literature on the neural correlates between two widely used measures. Our investigation of the neural links between parental bonding styles and social dominance has elucidated the external influences during the development of attitudes in childhood.

## 6. Conclusions/Summary

In summary, our findings suggest that increased social dominance and greater perceived parental protection during childhood are associated with increased amygdala GMV, and a negative association between perceived parental care with amygdala GMV. While behavioural correlations between PBI and SDO were weak and non-significant, the neuroimaging analyses revealed meaningful associations at the neural level, suggesting that parental bonding styles during childhood may have an influence on different regions of the brain associated with one’s social dominance, and potentially affect SDO development. These findings highlight the importance of multi-level analysis in understanding complex psychosocial phenomena. This study provides preliminary evidence for the possibility of a correlation between parenting and its effect on the development of prejudice at the neural level. However, it is important that we continue to build on the foundational work of other researchers and further investigate how these two concepts may be interlinked.

## Figures and Tables

**Figure 1 brainsci-15-01147-f001:**
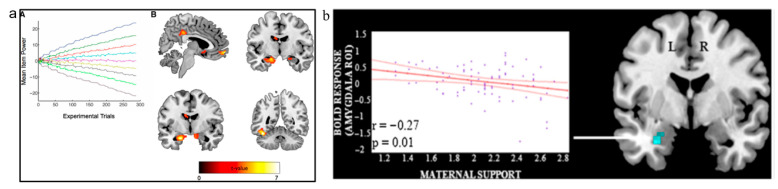
Relevant studies on neural activity associated with parental bonding styles. (**a**) good correlation between neural activity in the amygdala, hippocampus, ventromedial prefrontal cortex (vMPFC) [27]. (A) depicts an illustrative plot showing the evolution of the mean (expected) power values for each of the nine hierarchical positions throughout the experiment. Experimental trials refer to the number of task repetitions used to establish stable activation patterns; mean item power refers to the average strength of neural signal correlations across items. (B) The figure demonstrates that these brain regions (i.e., the bilateral amygdala (top right), vmPFC and posterior cingulate cortex (top left), bilateral anterior hippocampus (bottom left), and region proximate to the fusiform face area (bottom right)) show coordinated activity during the processing of social hierarchical information, supporting their role in social cognition. (**b**) significant voxel-wise negative correlation between good maternal parenting and left amygdala activation [5]. This indicates that adolescents who experienced more positive maternal behaviours showed reduced amygdala reactivity, suggesting that supportive parenting may modulate emotional processing in this region. Both (**a**,**b**) were retrieved with permission from their respective authors.

**Table 1 brainsci-15-01147-t001:** A classification of parental bonding styles based on PBI scores.

Parental Bonding Quadrants In Addition to Generating Care and Protection Scores for Each Scale, Parents Can be Effectively “Assigned” to One of Four Quadrants:	
“affectionless constraint” = high care and high protection	“affectionless control” = high protection and low care
“optimal parenting” = high care and low protection	“neglectful parenting” = low care and low protection
Assignment to “high” or “low” categories is based on the following cut-off scores:	
For mothers, a *care* score of 27.0 and a *protection* score of 13.5. For fathers, a *care* score of 24.0 and a *protection* score of 12.5.	

**Table 2 brainsci-15-01147-t002:** Correlation of SDO to PBI and individual PBI subscales.

	SDO	PBI	PBI_Care_	PBI_Protection_
SDO	-	−0.10	0.15	−0.11

Significance level is at *p* < 0.05 (2-tailed).

**Table 3 brainsci-15-01147-t003:** A summary of the association between the bilateral amygdala with PBI and SDO.

					MNI Coordinates		
Anatomical Area	Measure	Side	Direction of Association	P***_uncorr_*** (*p* < 0.63)	x	y	z
Amygdala	SDO	L	+	0.09	−46	−2	−20
		R		0.16	38	14	−14
	PBI_Care_	L	-	0.006	−45	−15	−10
		R		0.001	34	18	−20
	PBI_Protection_	L	-				
		R	+	0.004	30	20	−27

## Data Availability

This project was pre-registered on the open-science framework (OSF) at http://osf.io/9aebg (accessed on 7 September 2025). All raw neuroimaging data can be found in the DR-NTU repository for both the first (doi:10.21979/N9/H0CPDV) and second (doi:10.21979/N9/BFZ26X) recruitment phases. The datasets generated and/or analyzed during the current study are available in DR-NTU (Data): CUET; https://researchdata.ntu.edu.sg/dataset.xhtml?persistentId=doi:10.21979/N9/H0CPDV (accessed on 7 September 2025), VBM preprocessed Files Study 1; https://researchdata.ntu.edu.sg/dataset.xhtml?persistentId=doi:10.21979/N9/6QVJ9Q (accessed on 7 September 2025), VBM Preprocessed Files study 2; https://researchdata.ntu.edu.sg/dataset.xhtml?persistentId=doi:10.21979/N9/NUQAHW (accessed on 7 September 2025).

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
