# Peer review of "Prejudice Formation in Childhood: How Parental Bonding Can Affect Social Dominance Orientation"

_brainsci, 2025, doi:10.3390/brainsci15111147_

Round 1

Reviewer 1 Report

Comments and Suggestions for Authors

Dear Authors,

Overall, this research article is quite interesting. The area chosen is in need of time. Parenting is changing in this digital era. However, I have a few queries with respect to the methods and the results obtained. Kindly check the questions below.

In Figures 1a and b, the correlation images can be of better quality. 

In Figure 1a, the correlation graph displays experimental trials and the mean item power. What are these parameters?

Although the authors obtained permission to use the figures, providing an explanation would make it easier for readers to understand the concept.

In the introduction before the methods, the authors used the same sentence that was in the abstract. 

Kindly state the hypothesis clearly.

In methodology, the authors mentioned that 82 participants were recruited for the study in two phases; however, the phase data show a different proportion of participants. Kindly clarify.

What is the total study population in both phase I and phase II? How many male and female participants were involved?

Participants in the second phase were local Singaporean Chinese adults. What is the age group? On what basis were these participants selected for the study? Are there any specific inclusion criteria for the two different groups?

The authors didn't mention the exclusion criteria.

The PBI scale measures parental bonding styles during the first 16 years of life. However, the authors selected participants from the age group of 21 and above. Why?

How were the cut-off scores determined for this scoring?

I am unable to recover the dataset mentioned by the authors for verifying the results. Few results are available; if the authors incorporate them into the manuscript, it will be helpful to access.

Author Response

Comment 1: In Figures 1a and b, the correlation images can be of better quality. 

Response 1: Thank you for pointing this out. We have revised Figures 1a and 1b to be larger to improve readability.

Comment 2: In Figure 1a, the correlation graph displays experimental trials and the mean item power. What are these parameters?

Response 2: We apologise for the lack of explanation. “Experimental trials” refer to the number of task repetitions used to establish stable activation patterns, while “mean item power” refers to the average strength of neural signal correlations across items. We have clarified this in the figure legend for Figure 1a.

Comment 3: Although the authors obtained permission to use the figures, providing an explanation would make it easier for readers to understand the concept.

Response 3: We agree and have expanded the figure legends to include concise explanations of the main findings, ensuring that readers unfamiliar with the original studies can interpret the figures.

Comment 4: In the introduction before the methods, the authors used the same sentence that was in the abstract.

Response 4: Thank you for this observation. We have revised the introduction to avoid repetition and improve the flow.

Comment 5: Kindly state the hypothesis clearly.

Response 5: The hypotheses are now clearly stated at the end of the Introduction section and numbered (1–5).

Comment 6: In methodology, the authors mentioned that 82 participants were recruited for the study in two phases; however, the phase data shows a different proportion of participants. Kindly clarify.

Response 6: We have clarified this point in the Methods section. Ninety-one (46 females, age range = 21 - 42) participants were recruited for this study in 2 phases. 56 (27 females) participants (M = 23.05, SD = 1.30) recruited for the first phase were Singaporean Chinese students from Nanyang Technological University (NTU). In the second recruitment phase, identical inclusion and exclusion criteria were used. The only difference for this phase was the target sample, which was middle-aged adults from the community.There were 35 (19 females) participants (M = 31.59, SD = 6.48) in the second phase.

Comment 7: What is the total study population in both phase I and phase II? How many male and female participants were involved?

Response 7: As noted above, the breakdown is: Phase I (27 females, 29 males), Phase II (19 females, 16 males). This information has been added explicitly to the Methods section.

Comment 8: Participants in the second phase were local Singaporean Chinese adults. What is the age group? On what basis were these participants selected for the study? Are there any specific inclusion criteria for the two different groups?

Response 8: We have clarified that Phase II participants were also aged 21–42 years. Both phases shared the same inclusion criteria (Chinese ethnicity, English-speaking, right-handed, no psychiatric/neurological illness, etc.). Additional screening ensured participants had not travelled overseas for more than two months prior to scanning.

Comment 9: The authors didn't mention the exclusion criteria.

Response 9: Inclusion and exclusion criteria (history of psychological/neurological disorders, pregnancy, substance use, significant overseas travel) have been detailed in the Methods section.

Comment 10: The PBI scale measures parental bonding styles during the first 16 years of life. However, the authors selected participants from the age group of 21 and above. Why?

Response 10: We chose participants aged 21+ to ensure stable retrospective reporting of childhood experiences and to avoid confounds from ongoing adolescent development. This rationale is now added to the Methods section.

Comment 11: How were the cut-off scores determined for this scoring?

Response 11: The cut-offs were derived from the original PBI validation study (Parker et al., 1979) and subsequent replication work (Wilhelm & Parker, 1990). These references and thresholds (care: 27.0 mothers/24.0 fathers; protection: 13.5 mothers/12.5 fathers) are clearly described in the manuscript (see Table 1).

Comment 12: I am unable to recover the dataset mentioned by the authors for verifying the results. Few results are available; if the authors incorporate them into the manuscript, it will be helpful to access.

Response 12: We will ensure that all data are publicly accessible in DR-NTU with permanent DOIs (CUET; https://researchdata.ntu.edu.sg/dataset.xhtml?persistentId=doi:10.21979/N9/H0CPDV, VBM preprocessed Files Study 1; https://researchdata.ntu.edu.sg/dataset.xhtml?persistentId=doi:10.21979/N9/6QVJ9Q, VBM Preprocessed Files study 2; https://researchdata.ntu.edu.sg/dataset.xhtml?persistentId=doi:10.21979/N9/NUQAHW).

Reviewer 2 Report

Comments and Suggestions for Authors

  1. The manuscript states five clear hypotheses. However, the reported results in the self-report data section (weak, non-significant correlations and non-significant regression) directly contradict Hypothesis 1 and 2. The neurological findings for the amygdala (Hypotheses 3-5) are presented, but the discussion does not adequately address this stark discrepancy between the behavioural and neural levels of analysis. The authors should explicitly discuss this divergence, exploring potential reasons (e.g., limitations of self-report, the possibility that neural correlates are more robust or precede conscious attitudes) and temper the conclusions drawn from the hypotheses accordingly.
  2. The use of an uncorrected voxel-wise threshold of p < 0.01 with a 20-voxel extent for the whole-brain and ROI analyses is notably lenient and increases the risk of Type I errors (false positives). This is particularly concerning for the whole-brain analysis, which is exploratory. The authors should justify this threshold choice more thoroughly and report if any findings survive more stringent correction (e.g., FWE or FDR correction). Furthermore, for the ROI analysis, the p-values reported (e.g., p < 0.63) are unclear; this appears to be the uncorrected p-value, but the notation is confusing and should be clarified.
  3. The positive association between PBI Protection and amygdala GMV is interpreted through the lens of "negative connotations of increased parental control." However, the PBI's overprotection subscale can sometimes reflect anxious involvement rather than purely negative control. The discussion would benefit from a more nuanced interpretation of this subscale and its potential dual nature (concern vs. control) and how that might relate to brain development and SDO.
  4. The authors correctly identify the homogeneity of the sample (all of Chinese ethnicity in Singapore) as a limitation. This should be emphasized more strongly in the discussion. The cultural specificity of parenting styles (e.g., the mentioned patriarchal norms) and their perception could significantly influence the results. The findings, while valuable, may not be generalizable to Western or other cultural contexts. Future work should explicitly aim for cross-cultural replication.
  5. The regression equation presented on page 7-8 (SDO = 88.42 + 0.182B; – 0.11 B2) is unclear. What do 'B' and 'B2' represent? This should be explicitly defined (presumably PBICare and PBIProtection).

    For the ROI analysis, it would be good practice to also report results using a small volume correction (SVC) within the anatomically defined amygdala ROIs, as this is a more standard approach for testing a priori hypotheses.

Author Response

Comment 1: The manuscript states five clear hypotheses. However, the reported results in the self-report data section (weak, non-significant correlations and non-significant regression) directly contradict Hypothesis 1 and 2. The neurological findings for the amygdala (Hypotheses 3-5) are presented, but the discussion does not adequately address this stark discrepancy between the behavioural and neural levels of analysis. The authors should explicitly discuss this divergence, exploring potential reasons (e.g., limitations of self-report, the possibility that neural correlates are more robust or precede conscious attitudes) and temper the conclusions drawn from the hypotheses accordingly.

Response 1: We thank the reviewer for highlighting this. Specifically, we note that self-report measures may be limited by recall biases or social desirability, whereas neural correlates may reflect underlying processes that are not captured behaviourally. This discrepancy highlights the importance of a multi-level approach (self-report and neural) in studying social dominance, and we have added in a new paragraph in the discussions segment, and revised our limitations and conclusion segments accordingly.

Comment 2: The use of an uncorrected voxel-wise threshold of p < 0.01 with a 20-voxel extent for the whole-brain and ROI analyses is notably lenient and increases the risk of Type I errors (false positives). This is particularly concerning for the whole-brain analysis, which is exploratory. The authors should justify this threshold choice more thoroughly and report if any findings survive more stringent correction (e.g., FWE or FDR correction). Furthermore, for the ROI analysis, the p-values reported (e.g., p < 0.63) are unclear; this appears to be the uncorrected p-value, but the notation is confusing and should be clarified.

Response 2: We appreciate this important point. We have clarified that these are uncorrected p-values, and the notation has been revised to improve clarity throughout the Results section. As this was an exploratory study, this threshold was chosen to minimise Type II error while still applying a cluster-extent requirement. We recognise the increased risk of false positives and have now explicitly acknowledged this limitation in the Discussion. Then notation has also been revised for the results section (Whole Brain Analysis and ROI Analysis).

Comment 3: The positive association between PBI Protection and amygdala GMV is interpreted through the lens of "negative connotations of increased parental control." However, the PBI's overprotection subscale can sometimes reflect anxious involvement rather than purely negative control. The discussion would benefit from a more nuanced interpretation of this subscale and its potential dual nature (concern vs. control) and how that might relate to brain development and SDO.

Response 3: We agree. We have revised the Discussion and Limitations to include the dual interpretation of the overprotection subscale, which can reflect both anxious concern and controlling behaviours.

Comment 4: The authors correctly identify the homogeneity of the sample (all of Chinese ethnicity in Singapore) as a limitation. This should be emphasized more strongly in the discussion. The cultural specificity of parenting styles (e.g., the mentioned patriarchal norms) and their perception could significantly influence the results. The findings, while valuable, may not be generalizable to Western or other cultural contexts. Future work should explicitly aim for cross-cultural replication.

Response 4: We have strengthened the Limitations section to highlight the cultural specificity of the Singaporean Chinese sample, noting that parental bonding perceptions are culturally embedded. We will also recommend future cross-cultural replication.

Comment 5: The regression equation presented on page 7-8 (SDO = 88.42 + 0.182B; – 0.11 B2) is unclear. What do 'B' and 'B2' represent? This should be explicitly defined (presumably PBICare and PBIProtection).

Response 5: We have revised the regression model to clearly define terms: B1 = PBICare, B2 = PBIProtection. This correction appears in the Results section.

Comment 6: For the ROI analysis, it would be good practice to also report results using a small volume correction (SVC) within the anatomically defined amygdala ROIs, as this is a more standard approach for testing a priori hypotheses.

Response 6: We agree that small volume correction (SVC) is a more standard approach for ROI analyses. However, as this study was completed several years ago and we no longer have access to the raw imaging data, we are unable to re-run the analyses. At the time of analysis, we used an uncorrected voxel-wise threshold with an extent threshold, consistent with other exploratory VBM studies in small samples. We acknowledge this as a limitation in the manuscript and note that future research should adopt SVC or stricter correction methods to strengthen the robustness of findings.